# Morphological correlates of synaptic protein turnover in the mouse brain

Fengxia Li[1,2], Julius N Bahr[2,3,4] , Felicitas A-L Bierth[2,3,5], Sofiia Reshetniak[1] , Christian Tetzlaff[1], Eugenio F Fornasiero[1], Carolin Wichmann[2,3], Silvio O Rizzoli[1,2]

Synaptic proteins need to be replaced regularly, to maintain function and to prevent damage. It is unclear whether this process, known as protein turnover, relates to synaptic morphology. To test this, we relied on nanoscale secondary ion mass spectrometry, to detect newly synthesized synaptic components in the brains of young adult (6 mo old) and aged mice (24 mo old), and on transmission electron microscopy, to reveal synapse morphology. Several parameters correlated to turnover, including pre- and postsynaptic size, the number of synaptic vesicles and the presence of a postsynaptic nascent zone. In aged mice, the turnover of all brain compartments was reduced by ~20%. The turnover rates of the pre- and postsynapses correlated well in aged mice, suggesting that they are subject to common regulatory mechanisms. This correlation was poorer in young adult mice, in line with their higher synaptic dynamics. We conclude that synapse turnover is reflected by synaptic morphology.

## Introduction

Aged neuronal proteins are replaced regularly, on a time scale of days to weeks, to ensure the long-term maintenance of function, as well as plasticity-related changes (Cohen et al, 2013; Dörrbaum et al, 2018; Fornasiero et al, 2018; Dörrbaum et al, 2020). The turnover process seems to be carefully controlled, resulting in turnover rates that correlate to many fundamental parameters, from the compartments in which the proteins reside (Price et al, 2010) to the structure of the respective proteins, or even to their amino acid sequences (Mandad et al, 2018). Importantly, protein turnover is also connected to the overall levels of neuronal activity, and to the function of the respective synapses (Bingol & Sheng, 2011; Schanzenbächer et al, 2016; Heo et al, 2018; Schanzenbächer et al, 2018). For example, increasing synaptic activity results in higher rates of exo- and endocytosis in the presynapse, which induces the accelerated aging and degradation of synaptic vesicles (Truckenbrodt et al, 2018). This is balanced by increased rates of delivery of synaptic vesicle proteins to the respective synapses (Jähne et al, 2021).

It is unclear whether protein turnover also relates to synaptic morphology and overall appearance. The connection between turnover and synaptic function implies that such a relation is not unlikely since synaptic morphology is closely attuned to synaptic function. For example, the fraction of releasable (recycling) synaptic vesicles scales inversely with synapse size for Schaffer collateral synapses (Rose et al, 2013), implying that large synapses contain numerous inactive (nonrecycling) vesicles, whereas smaller ones use most of their vesicles. At identical levels of activity, the larger synapses would turn over their vesicles more slowly than small ones since activity accelerates the vesicle degradation (Truckenbrodt et al, 2018).

Some morphological indicators of turnover may also exist for the postsynaptic compartment. For example, stimuli that induce long-term potentiation also result in the formation of nascent zones, which are postsynaptic areas that exhibit a postsynaptic density, but lack presynaptic vesicles on the other (presynaptic) side of the synapse (Bell et al, 2014). This implies that nascent zones are strong indicators of synaptic dynamics, which presumably demonstrate synaptic remodeling, which may require the synthesis of new synaptic proteins (albeit significant turnover may take place without such synthesis, simply by local remodeling of synaptic elements as the vesicle pools).

In spite of these indications, no morphological markers of synaptic turnover have actually been demonstrated, due to technical difficulties. Typically, cells, tissues or animals are fed with isotopically labeled molecules, as amino acids. Their incorporation into cellular components is then analyzed by mass spectrometry (e.g., stable isotope labeling by amino acids in cell culture, SILAC, [Ong et al, 2002]). Most analyses of cellular turnover are performed by fluid-based mass spectrometry approaches, losing cellular or synaptic morphology.

To address this problem, we turned to an imaging mass spectrometry approach, nanoscale secondary ion mass spectrometry (nanoSIMS; [Lechene et al, 2006]). NanoSIMS functions by irradiating the samples of interest with a primary ion beam (e.g., a $Cs^+$ beam),

[1]Department of Neuro- and Sensory Physiology, University Medical Center Göttingen, Göttingen, Germany   [2]Center for Biostructural Imaging of Neurodegeneration, University Medical Center Göttingen, Göttingen, Germany   [3]Molecular Architecture of Synapses Group, Institute for Auditory Neuroscience and InnerEarLab, University Medical Center Göttingen, Göttingen, Germany   [4]Göttingen Graduate Center for Neurosciences, Biophysics and Molecular Biosciences (GGNB), University of Göttingen, Göttingen, Germany   [5]Molecular Medicine Bachelor Programme, University Medical Center Göttingen, Göttingen, Germany

Correspondence: srizzol@gwdg.de; carolin.wichmann@med.uni-goettingen.de

which causes the release of secondary particles from the irradiated surface. The released particles include ionized components, which are revealed by mass spectrometry detectors. For example, if cells are exposed to amino acids containing $^{13}C$ isotopes, their incorporation into proteins can be followed by nanoSIMS, focusing on the detection of $^{13}C$, as a marker for regions in which protein turnover took place. The spatial resolution of this technique, around 50–100 nm in the lateral plane, and 5–10 nm in the axial direction (Lechene et al, 2006; Saka et al, 2014), is sufficient for the analysis of synaptic components, albeit a second technology is needed to reveal the nature of the different organelles. Here, we correlated nanoSIMS with transmission electron microscopy (TEM), which provides excellent cellular and organelle identification (as we have shown in the past; [Lange et al, 2021; Michanski et al, 2023]). Importantly, we would like to point out that our work can only refer to synaptic changes that are due to the appearance of newly synthesized material. Synaptic changes that only refer to rearrangements of existing molecules and organelles, for example, through the exchange or sharing of such elements between synapses, will not be visualized in our experiments.

Using correlated TEM-nanoSIMS, we analyzed mice that were fed an isotope-enriched diet, before analyzing their brains and synapses. Focusing on three different brain areas, the hippocampal CA1 and dentate gyrus (DG) areas, as well as the piriform cortex, we found that several morphological features of synapses relate to their turnover status, from size and synaptic vesicle number to the presence of a nascent zone. Importantly, these features were similar for the different brain regions, and were detected in both young adults and aged mice. Interestingly, this analysis also pointed to a correlation between the turnover of the pre- and postsynaptic compartments, which was especially strong for the aged mice. This implies that the pre- and postsynapses, which belong to different cells, and should, in principle, have independent turnover rates, are subject to common regulation mechanisms, in terms of turnover.

# Results

## A correlative nanoSIMS-TEM analysis of synaptic components

To analyze the turnover of synaptic proteins, several studies have relied on isotopically labeled amino acids, like lysine. This approach results in the incorporation of this label into proteins with high fidelity, but only provides a relatively small signal since lysine only makes up ~6% of the amino acids in the mouse brain proteome. An alternative is the use of a diet in which all nitrogen atoms are provided in the form of isotopic $^{15}N$ (Wu, 2004; McClatchy et al, 2007). This results in the strong isotopic labeling of all proteins, and also of other nitrogen-containing cellular components, as nucleic acids and lipids. The subsequent processing, which includes fixation, ethanol dehydration and plastic embedding, results in the loss of small metabolites (Lork et al, 2024), and a very large fraction of the lipids (e.g. [Morgan & Huber, 1967]), implying that the remaining $^{15}N$ signal is mostly derived from proteins and/or nucleic acids. The latter are poorly fixed, and will be lost to a higher extent than proteins from the samples, during fixation and plastic embedding (e.g. [Urieli-Shoval

et al, 1992]). The samples were sectioned to 110 nm in thickness, were imaged in TEM, and were then subjected to the nanoSIMS analysis (Fig 1A). The images were subsequently aligned, enabling the detection of specific cells, synapses and organelles (Lange et al, 2021; Michanski et al, 2023). The molecular turnover was then calculated by calculating the isotopic ratio between "new" and "old" nitrogen atoms, which are measured in the form of $^{12}C^{15}N^-$ and $^{12}C^{14}N^-$ ions, respectively (denoted, for simplicity, as $^{15}N$ and $^{14}N$ in the rest of this work). The animals were fed with this isotopic diet for 21 d, ensuring that the turnover of their proteins can be analyzed at steady-state point, rather than during periods of rapid changes (Fornasiero et al, 2018). As shown in Fig 1B, the turnover levels are heterogeneous, with specific cellular compartments exhibiting $^{15}N/^{14}N$ ratios (indicating the ratio between new and old synaptic components) up to 10-fold higher than others.

To analyze this heterogeneity in detail, we targeted the following compartments, in addition to the whole area depicted in each image, which served as our baseline: pre- and postsynapses, as well as their most obvious morphological features, the active zone, the vesicle cluster and the postsynaptic density (PSD); myelin, along with the axons it surrounds; mitochondria; neurites (with unclear identity). We analyzed the size or morphology of these elements in 2D EM images, selecting the respective areas by hand drawing. Their turnover values were then analyzed in the corresponding nanoSIMS images (Fig S1). We first analyzed young adult animals (Fig 2). Overall, the general turnover of axons and other neurites followed that of the respective brain areas (Fig 2A and B). Myelin (Fig 2A, Zoom 2), as expected from its slow protein turnover (Savas et al, 2012; Toyama et al, 2013), exhibited low average values for the $^{15}N/^{14}N$ ratio (Fig 2B). In contrast, synapses (Fig 2A, Zoom 1) contained higher levels of newly synthesized components, with the presynaptic active zone and PSD being the most turnover-active elements in the brain, is in line with their role in plasticity (Fig 2B). Mitochondria (Fig 2A, Zoom 3), did not, as a population, exhibit particularly high or low turnover values (Fig 2B).

A similar situation was observed for aged mice (Fig 3A). Mice of 24 mo age are expected to have a slower overall protein turnover, compared with young adults (Kluever et al, 2022). This was indeed observed, with the average $^{15}N/^{14}N$ ratios being lower by ~19% (see values in Figs 2 and 3B), a value that compares well with the ~20% observed in proteomics analyses (Kluever et al, 2022). The relative turnover rates of all of the analyzed compartments (Fig 3) remained remarkably similar to those from young adult mice (Fig 2), leading to the same observations, which include a relatively lower turnover of myelin, or the high turnover of pre- and postsynaptic components.

To analyze the synapses in more detail, we normalized the different turnover ratios of each compartment to the respective whole brain areas, for both the young and old mice. For the postsynapse, the active zone and the PSD, we observed no substantial differences between young and aged mice: these components turn over faster than the brain as a whole, and the differences between the synapses and the whole brain area are similar for the young and aged mice. For the presynapse, however, turnover in the aged mice was slower, when normalized to the whole brain area, than for young mice, suggesting that presynaptic turnover slows down during aging even more than

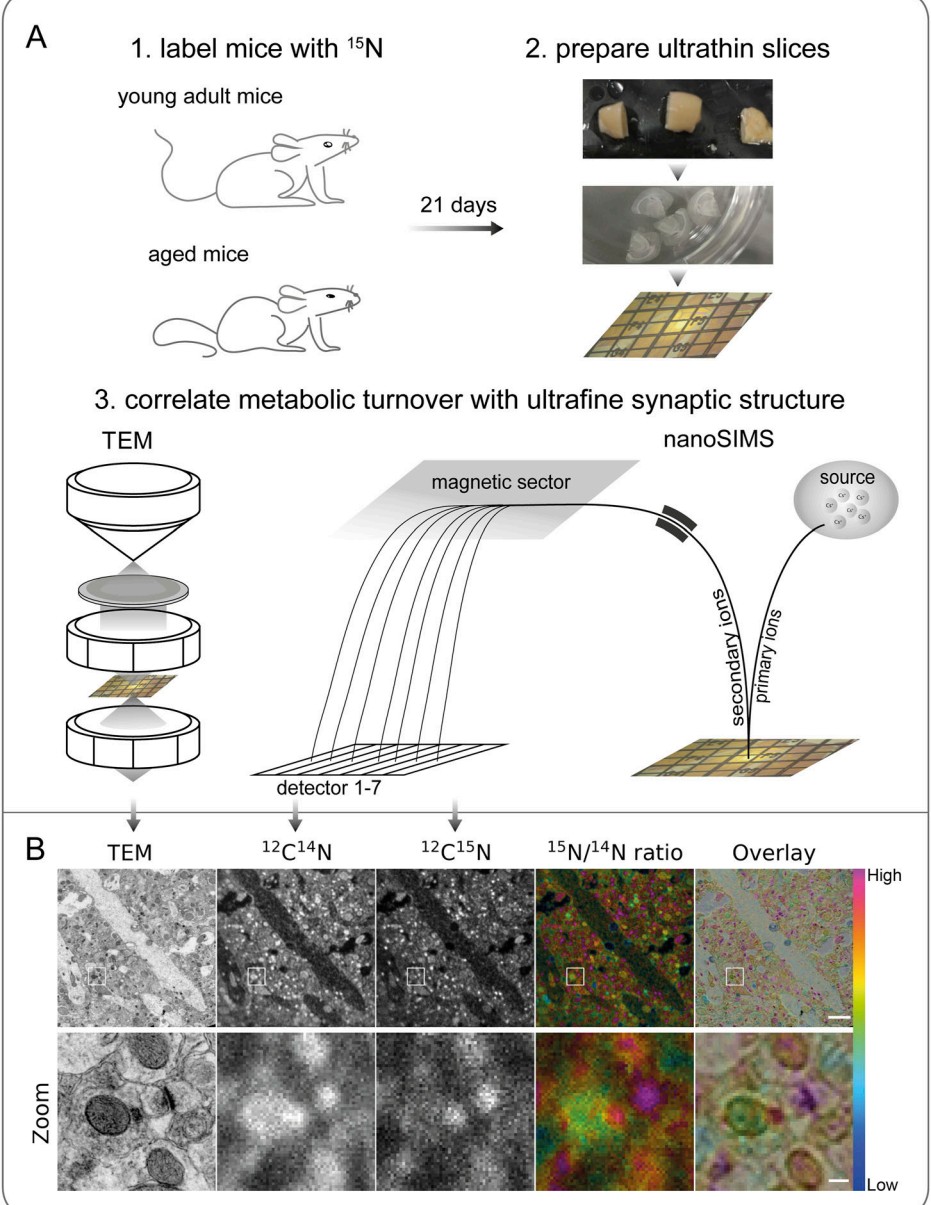

**Figure 1. TEM-nanoSIMS analysis of mouse brains.**
**(A)** Schematic view of the experimental workflow. (A1) Young adult (6 mo old) and aged (24 mo old) mice were fed with food containing only $^{15}N$ for 21 d. (A2) The animals were perfusion-fixed, the brains were then extracted, and the desired regions of interest were targeted and vibratome slices prepared, plastic-embedded and ultrathin-sectioned for TEM imaging, before being placed on mesh grids. (A3) Sections were first imaged by TEM, and then by nanoSIMS, to reveal brain structure as well as chemical maps, which can be correlated for a structure-guided turnover analysis. **(B)** The upper panel shows images from a representative area of the CA1 region in a young adult mouse. $^{12}C^{14}N$ and $^{12}C^{15}N$ images generated by nanoSIMS are used to calculate the $^{15}N/^{14}N$ ratio image, which is then correlated to the respective TEM image. The bottom panel shows an enlarged view of the area outlined in the top panel. Scale bars: upper panel 2 $\mu m$, bottom panel 200 nm.

that of the brain as a whole. While the difference is small (Fig 3C), it is significant, and is observed in all of the three brain areas we investigated (see also detailed descriptions of findings in the DG and cortex; see Figs S2A and B, S3A and B, S4A and B, and S5A and B).

### Morphological parameters in relation to synaptic turnover

We next focused on the relation between pre- and postsynaptic morphology and protein turnover. As morphological parameters require dedicated manual analysis for each object, we did not analyze the entire dataset (of more than 5,000 synaptic and non-

synaptic objects), but measured the different morphological parameters in subsets of brain components (e.g., synapses) exhibiting the highest or lowest turnover ratios. This analysis is therefore able to indicate whether components with very high or very low turnover exhibit specific morphological characteristics.

Synapse size is an evident characteristic, which has been linked to synaptic weight and function (Humeau & Choquet, 2019). We therefore analyzed the area of the different synapse sections (while acknowledging that this is an imperfect parameter, with 3D volumes being much preferable; unfortunately, these are currently difficult to obtain in a TEM-SIMS configuration). Presynaptic size proved to have some correlation to turnover (Fig 4A and B), with larger

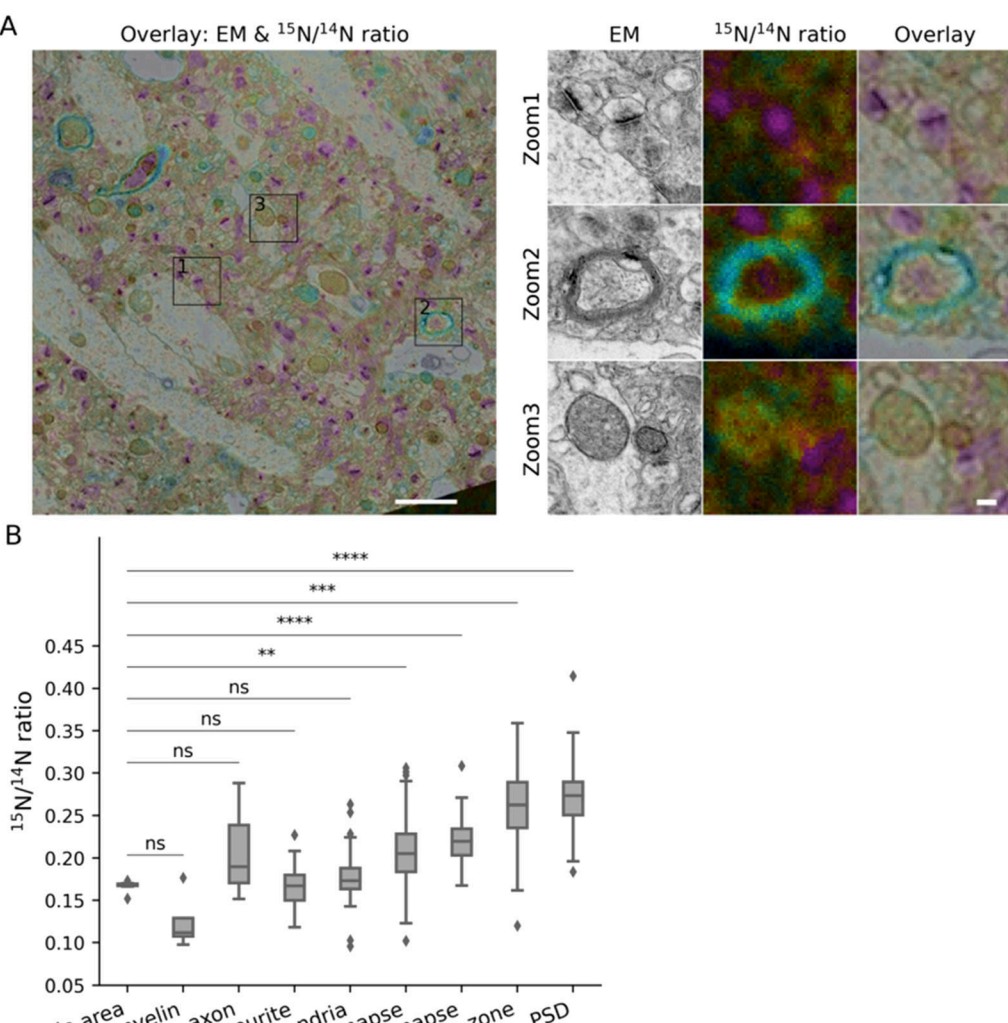

**Figure 2. Turnover analysis for different brain structures in the CA1 region of young adult mice.**
**(A)** Left panel: overlay of an example TEM image and its corresponding $^{15}N/^{14}N$ ratio image. Right panel: zoom areas, showing enlarged views of the regions indicated by squares in the left panel, depicting synapses (Zoom 1), myelin (Zoom 2) and a mitochondrion (Zoom 3). Scale bars: left panel 2 $\mu$m, right panel 200 nm. **(B)** Box-plot of $^{15}N/^{14}N$ ratios in the different brain structures (three different mice). Differences between the different structures and the whole brain area were tested using Mann-Whitney tests, followed by a Bonferroni multiple testing correction. The following Bonferroni-corrected $P$-values were obtained: myelin, $P = 1.0$ (n analyzed objects = 4); axon, $P = 1.0$ (n = 3); neurite, $P = 1.0$ (n = 46); mitochondria, $P = 1.0$ (n = 47); presynapse, $P = 0.0029$ (n = 161); postsynapse, $P = 7.0 \times 10^{-5}$ (n = 193); active zone, $P = 2.1 \times 10^{-4}$ (n = 110); PSD, $P = 6.0 \times 10^{-5}$ (n = 177). The box plots show the median (middle line), the lower and upper quartiles (box), the minimum and maximum numbers that are not outliers (the whiskers), and the outliers (calculated using the interquartile range).

synapses showing less turnover (tending to be metabolically older). The correlation became far stronger, and visible in all analyzed brain areas, when investigating the synaptic vesicle numbers in each section (Fig 4C), rather than just the presynaptic area (Fig 4B). The number of docked vesicles did not seem to correlate with turnover (e.g., for the active zone; Kruskal-Wallis test, $P > 0.3$).

For the postsynapse, the synapse area was a clear indicator of turnover, with large synapses showing slower turnover compared with small synapses (Fig 5A and B). The correlation to size is not artifactual because of it does not hold for other structures, as mitochondria (Fig S6A–C), neurites (Fig S7A and B), or neurites (Fig S8A and B), for which no such relations are expected. Mitochondria turnover seems to depend more on the respective mitochondria location, than on size, with presynaptic mitochondria being especially "old" (Fig S6A–C). The turnover of axons and neurites presumably reflects the metabolic states of the respective cells, rather than local phenomena that can be measured in TEM-nanoSIMS, and therefore no correlations to their size are expected. Finally, myelin thickness correlates to myelin turnover (Fig

S9A and B), again in agreement with the very slow turnover of myelin components (Meschkat et al, 2022). Interestingly, the relation to size does not provide any insight for the PSD (Fig 6A), whose area (Fig 6B) or length (Fig 6C) do not seem to relate to turnover. The turnover of the PSD relates much more closely to the presence of a nascent zone (Fig 6D), for both young and aged animals (Fig S10).

**The correlation between pre- and postsynaptic turnover**

In principle, the two synaptic compartments (pre- and postsynaptic) are derived from two independent neurons, and therefore do not inherently need to correlate to each other in terms of turnover, being subject to different metabolic regulation, from two different cells. Confirming this hypothesis, a previous analysis of the turnover of PSDs and the vesicle clusters suggests a very limited degree of correlation, at least in hippocampal cultured neurons (Jähne et al, 2021). To test this in living mice, we analyzed several pre- and postsynaptic pairs (Fig 7A–C). Interestingly, in the CA1 area the

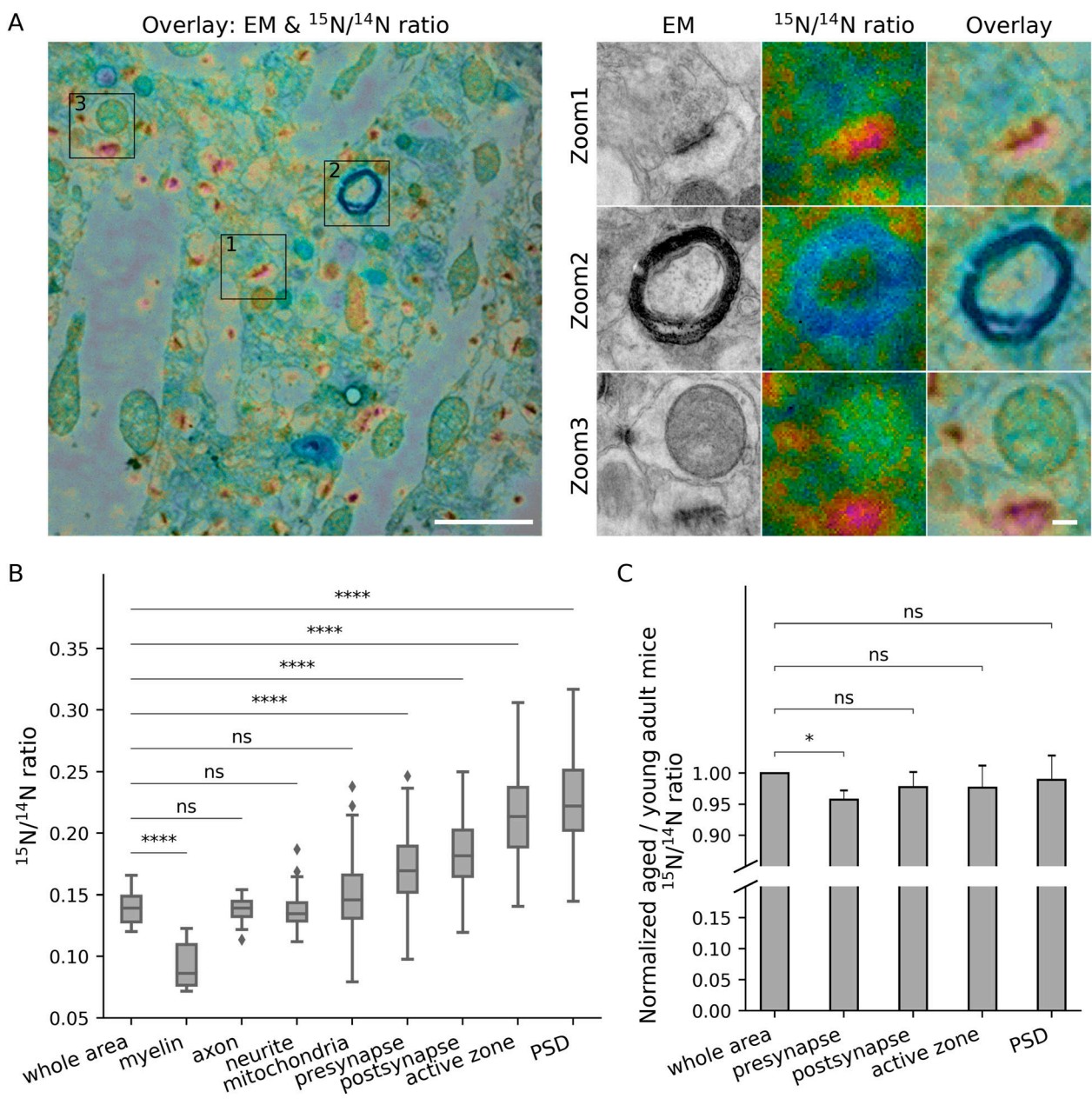

**Figure 3. Turnover analysis for different brain structures in the CA1 region of aged mice.**
**(A)** Left panel: overlay of an example TEM image and its corresponding $^{15}N/^{14}N$ ratio image. Right panel: zoom areas, showing enlarged views of the regions indicated by squares in the left panel, depicting a synapse (Zoom 1), myelin (Zoom 2) and a mitochondrion (Zoom 3). Scale bars: left panel 2 $\mu$m, right panel 200 nm. **(B)** Box-plot of $^{15}N/^{14}N$ ratios in the different brain structures (three different mice). Differences between the different structures and the whole brain area were tested using Mann-Whitney tests, followed by a Bonferroni multiple testing correction. The following Bonferroni-corrected *P*-values were obtained: myelin, $P = 5.9 \times 10^{-5}$ (n = 8); axon, $P = 1.0$ (n = 8); neurite, $P = 1.0$ (n = 30); mitochondria, $P = 0.98$ (n = 140); presynapse, $P = 2.1 \times 10^{-5}$ (n = 263); postsynapse, $P = 6.0 \times 10^{-9}$ (n = 351); active zone, $P = 3.0 \times 10^{-10}$ (n = 174); PSD, $P = 6.3 \times 10^{-11}$ (n = 276). **(C)** We normalized the isotopic ratios between the young and aged mice, by dividing all ratios to their overall, whole-organ averages, and then divided the values obtained in aged mice by those from young adult mice. This analysis indicates that the presynapse turns over significantly more slowly in the aged mice (N = 3 brain areas, cortex, CA1 and dentate gyrus). Differences were tested using *t* tests, followed by Bonferroni multiple testing correction (n = 3) with the following *P*-values: presynapse, $P = 0.030$; postsynapse, $P = 0.74$; active zone, $P = 1.0$; PSD, $P = 1.0$. The box plots show the median (middle line), the lower and upper quartiles (box), the minimum and maximum numbers that are not outliers (the whiskers), and the outliers (calculated using the interquartile range).

correlation between the pre- and postsynapse was not significant for young adult mice, but was relatively strong for aged mice (Fig 7D). A similar relation was found in the DG (Fig S11A–C), while the cortex synapses (Fig S12A–C) exhibited a strong correlation in both young and aged mice. Overall, this implies that the pre- and postsynaptic compartments tend to correlate, at least in the aged mice, suggesting a certain degree of co-regulation in terms of turnover.

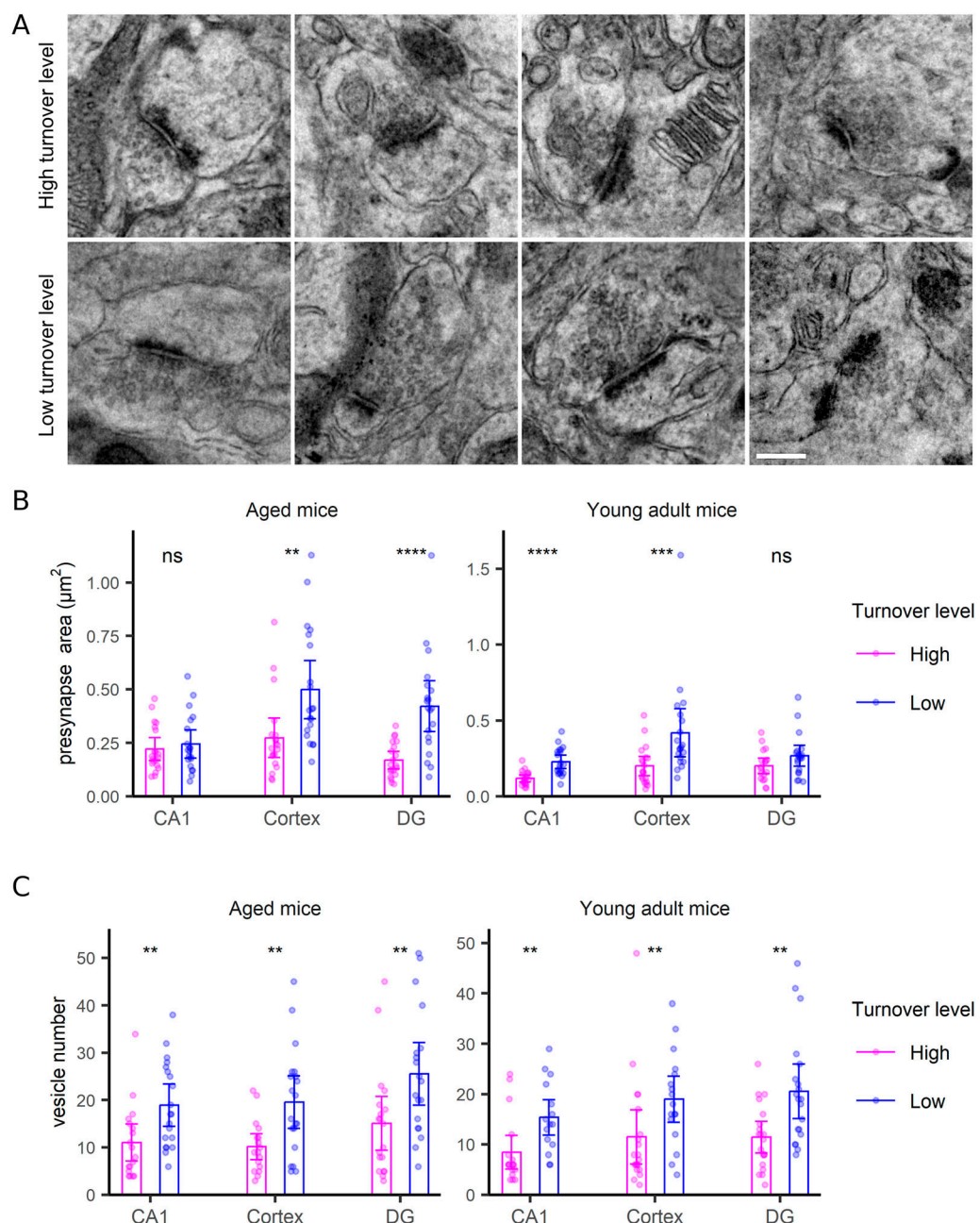

**Figure 4. Presynaptic morphology correlates with turnover.**
**(A)** TEM examples of presynaptic structures with the highest or lowest turnover rates. Scale bar: 250 nm. **(B, C)** Comparison of area (B) and vesicle numbers (C) in presynapses with the highest or lowest turnover levels, in the CA1, cortex and DG regions, in both aged and young adult mice. Differences were tested by the Wilcoxon test. $P$-values for Fig 4B are 0.49, 0.001, $2.1 \times 10^{-5}$, $1.6 \times 10^{-5}$, $3.7 \times 10^{-4}$, 0.13, respectively; for Fig 4C they are 0.0054, 0.0045, 0.0072, 0.0023, 0.0082, and 0.003, respectively. N = 15–20 synapses analyzed for each condition.

# Discussion

Overall, our work suggests that synaptic turnover correlates with several synaptic morphology parameters, including pre- and postsynaptic size, synaptic vesicle numbers and the presence of a synaptic nascent zone. Some of these correlations may be due to limitations posed by morphology on function and function-dependent turnover, as discussed in the Introduction, in relation to the use and turnover of synaptic vesicles. Other correlations are less obvious but may still relate to functional parameters, as discussed below.

The size of the presynapse correlates well to the number of synaptic vesicles (e.g. [Murthy et al, 2001]). The dynamics of the population of synaptic vesicles depend on their function because

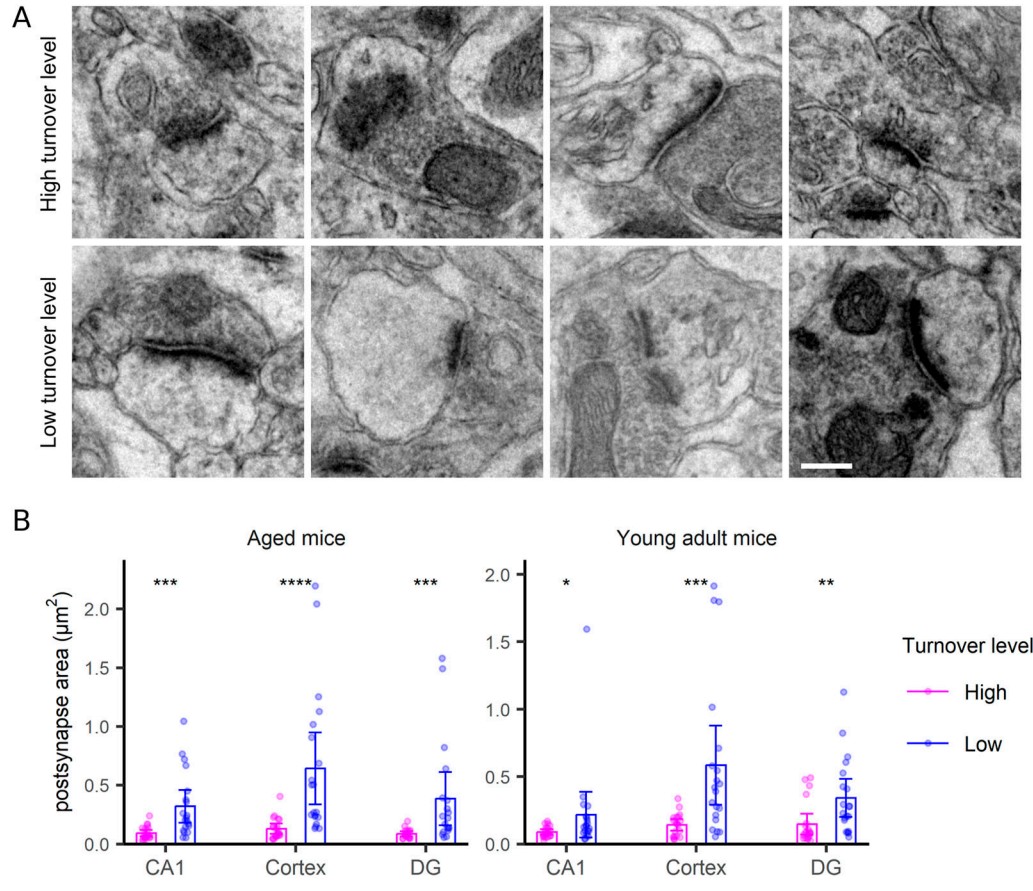

**Figure 5. Postsynaptic morphology correlates with turnover.**
**(A)** TEM examples of postsynaptic structures with the highest or lowest turnover rates. Scale bar: 250 nm. **(B)** Comparison of postsynaptic area in synapses with the highest or lowest turnover levels, in the CA1, cortex and DG regions, in both aged and young adult mice. Differences were tested by the Wilcoxon test. *P*-values for Fig 5B are $2 \times 10^{-4}$, $1.1 \times 10^{-6}$, $1.3 \times 10^{-4}$, 0.021, $2.6 \times 10^{-4}$ and 0.002, respectively. N = 18–20 synapses analyzed for each condition.

of vesicle protein turnover is directly linked to vesicle usage (Truckenbrodt et al, 2018). Evidence is rapidly accumulating in this field, suggesting that the synaptic vesicle cluster is organized according to liquid–liquid phase separation principles (Milovanovic & De Camilli, 2017; Milovanovic et al, 2018; Hoffmann et al, 2023), and that the vesicles, in turn, organize a multitude of synaptic components, from exo- and endocytosis cofactors to cytoskeletal components (Reshetniak & Rizzoli, 2021). Synaptic activity tends to dissipate synaptic vesicles, by modulating the vesicle-binding protein synapsin (e.g. [Milovanovic & De Camilli, 2017]), implying that synapses in which a large proportion of the vesicles undergo exo- and endocytosis will have less stable vesicle clusters, which presumably lose not only vesicles (to degradation, [Truckenbrodt et al, 2018]), but also other molecules, to diffusion outside of boutons. Large synapses, which contain proportionally lower levels of exo- and endocytosis cofactors (Wilhelm et al, 2014), will presumably have proportionally lower levels of recycling vesicles (Rose et al, 2013), and thus more stable vesicle clusters. These will retain vesicles and their associated molecules more strongly, implying that these molecules can become stabilized within the vesicle clusters, with the whole molecular assembly, and the presynapse area as a whole, becoming metabolically "old."

The correlation between large postsynaptic spines and slow turnover is, at first view, more difficult to explain. A first concept is that large size may simply relate to the synapse identity, with the larger ones belonging to neurons that turn over especially slowly. However, the fact that this correlation takes place in different brain regions, occupied by different types of neurons, implies that this solution is unlikely. A more likely hypothesis is linked to the observation that a dense and relatively homogeneous meshwork of actin fills most of the volume of synaptic spines (Eberhardt et al, 2022). Actin is a long-lived protein, whose isoforms have lifetimes that surpass the median values for brain proteins (Fornasiero et al, 2018). In contrast, most other postsynaptic components tend to be shorter-lived than the median of all brain proteins (Fornasiero et al, 2018), with the postsynaptic receptors being especially short-lived. This implies that synapses in which the volume occupied by actin is small, in relation to the PSD and the membranes containing synaptic receptors, will tend to have a more rapid turnover than synapses in which actin occupies a proportionally larger space. As the actin-occupied volume increases with the third power of the spine radius, while the membranes to which PSD molecules and receptors associate increase with the second power of this radius,

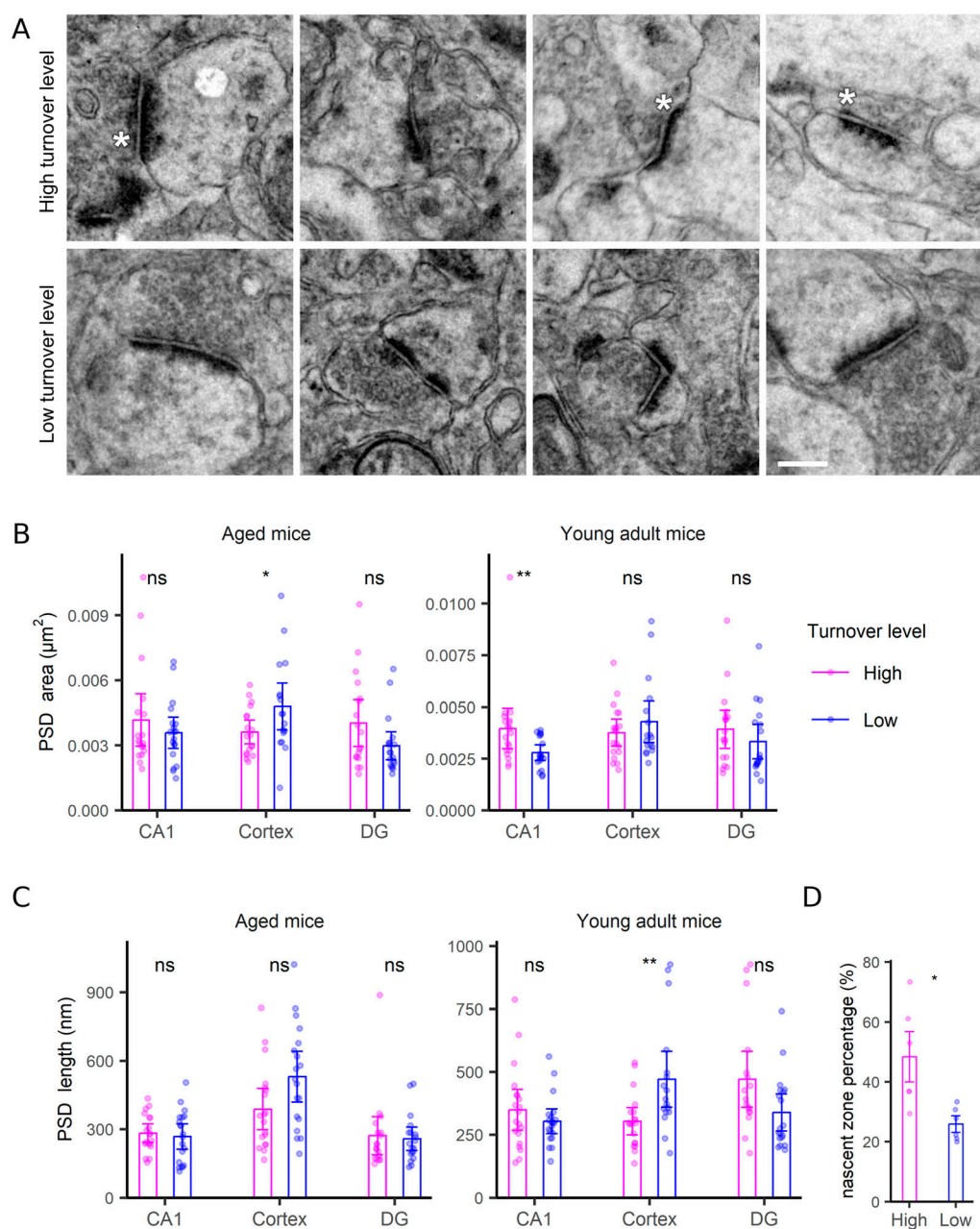

**Figure 6. Limited correlations between PSD morphology and turnover.**
**(A)** TEM examples of PSDs with the highest or lowest turnover rates. Asterisks indicate nascent zones. Scale bar: 250 nm. **(B, C)** Comparison of area (B) and PSD length (C) for PSDs with the highest or lowest turnover levels, in the CA1, cortex and DG regions, in both aged and young adult mice. The PSD area was defined as the area covered, in the synapse 2D profile, by the dense postsynaptic formation. The PSD length was defined as the longest axis of the PSD drawing. **(D)** An analysis of the PSDs exhibiting a nascent zone, expressed as % for each brain region (CA1, cortex, DG) and age (young, aged). Differences in Fig 6B–D were tested by the Wilcoxon test. *P*-values for Fig 6B are 0.89, 0.042, 0.092, 0.0053, 0.39, and 0.35, respectively. *P*-values for Fig 6C are 0.51, 0.056, 0.9, 0.47, 0.0043, and 0.059, respectively. *P*-value for Fig 6D is 0.013. **(C, D)** N = 17–20 synapses analyzed for each condition, for panel (C); N = 6 for panel (D) (three brain regions, across two ages).

it is clear that large spines will contain proportionally more actin, and will appear to be metabolically older.

Some of the correlations observed may also be due to a combination of physiological and technical issues. The correlation between small synaptic size and high turnover may reflect two points: (1) large synapses may simply be more stable, and smaller ones less stable; (2) edges of large PSDs, on large spines, may be

dynamic, and would appear as small postsynapses and PSDs on single sections.

The correlation between myelin thickness and turnover is not necessarily surprising, albeit, in principle, no such correlation needs to be encountered. As myelin is replaced by degrading "blocks" of membrane, followed by the incorporation of new myelin membranes (Meschkat et al, 2022), it is unclear whether the number

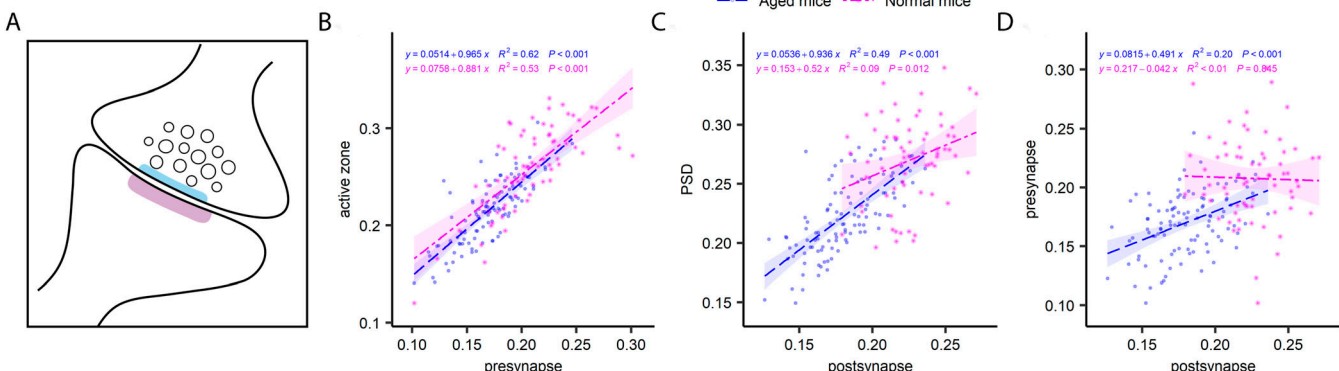

**Figure 7. The turnover of pairs of synapses in the CA1 region.**
**(A)** Cartoon of a typical synapse, indicating the presynapse, with its active zone (light blue) and presynaptic vesicles, as well as the postsynapse, containing the PSD (in magenta). **(B, C, D)** Comparisons of presynapse and active zone turnover (B), postsynapse and PSD turnover (C) and presynapse and postsynapse turnover (D). The graphs show $^{15}N/^{14}N$ ratios measured in paired structures. Although the first two comparisons show significantly correlations for both aged (blue) and young adult (pink) mice, the latter is only significant for the aged mice. P and $R^2$ values are indicated in the graphs.

of myelin layers (which is the parameter which influences myelin thickness the most) should be correlated to turnover. One solution to this question comes from considerations on the energetic efficiency of such a process, as myelin is costly to generate, and therefore thicker myelin sheets should balance their high production costs by high longevity (Georgiev & Rizzoli, 2023). Another consideration is that protein turnover seems to be correlated to the chances proteins have to become damaged by oxidation (Fornasiero et al, 2018), and oxidation should be reduced strongly by thicker myelin sheets, at least for the inner layers of this structure.

Nascent zones have been defined as regions in which the PSD is not mirrored by a vesicle-occupied active zone, on the presynaptic side (Bell et al, 2014). In principle, they seem to be plasticity-induced phenomena, which most likely relate to re-arrangements of pre-existing synaptic components, rather than the generation of new proteins. Therefore, they do not need to be inherently linked to synaptic turnover. Our observations (Fig 6) imply that nascent zones are found in synapses in which newly synthesized synaptic proteins were generated and/or recruited. Overall, we suggest that nascent zones could be regarded as indicators of highly dynamic synapses, in both functional and metabolic terms.

The turnover correlation between the pre- and postsynaptic compartments is more difficult to explain. In principle, the metabolism of the presynaptic compartment is controlled by presynaptic activity, as discussed in the previous paragraphs. The metabolism of the postsynaptic compartment should also respond to presynaptic activity, at least in what regards plasticity-imposed changes (e.g. [Jähne et al, 2021]). Therefore, a level of turnover correlation may simply be due to presynaptic activity controlling both synaptic compartments. However, this hypothesis does not consider age-related differences observed in the hippocampus. To take this finding into account, one would need to hypothesize that aged postsynapses respond mostly to presynaptic activity, while young ones also display presynapse-independent dynamics and/or instability. This proposal is in line with the reduced plasticity found in aged synapses (e.g. [Bergado & Almaguer, 2002]). More importantly, this hypothesis fits well with the observation that 90% of the newly generated spines tend to be lost over a few weeks in adult mice, while they are stable in aged mice (Mostany et al, 2013).

Overall, in young adult mice postsynapses may be relatively unstable, and may require frequent remodeling (implying substantial turnover), while in aged ones they tend to be stable, and would thus be more likely to align their turnover to that of the presynapse.

Overall, we conclude that synaptic morphology holds clues to turnover dynamics. To study the exact mechanisms of this relation, larger-scale studies, covering high numbers of synapses, in different metabolic conditions, would be required, perhaps through the use of automated (Schneider et al, 2012) segmentation approaches (e.g., Archit et al, 2023 *Preprint*). The results could then be used in the analysis of brain connectomes, providing an additional layer of information, in addition to the structural information they contain.

# Materials and Methods

### Mice

All animal experiments were approved by the local authority, the Lower Saxony State Office for Consumer Protection and Food Safety (Niedersächsisches Landesamt für Verbraucherschutz und Lebensmittelsicherheit).The mice used here (C57Bl/6JRj) of different ages (6 and 24 mo at the end of the labeling time) were purchased from Janvier labs and checked for absence of standard pathogens. For metabolic labeling animals were fed ad libitum with the $^{15}N$-labeled SILAM mouse diet (SILANTES), as previously described (Alevra et al, 2019).

### Animals (EM, NanoSIMS)

Each animal was deeply anaesthetized using $CO_2$ and was then intracardially perfused with 300 ml 0.1 M PBS (pH 7.4, P4417; Sigma-Aldrich) followed by 200 ml cold 4% PFA (pH 7.4, 0335.1; Carl Roth) in PBS. Subsequently, the brains were carefully removed from the skull, cut in the two hemispheres and incubated overnight in 2% PFA, 2.5% glutaraldehyde (G7651; Sigma-Aldrich) in 0.1 M cacodylate buffer, pH 7.2, and kept afterward in PBS.

## Vibratome sectioning

The vibratome sectioning was performed in ice-cold PBS. The brains were first trimmed coronally and the cerebellum as well as the olfactory bulb was removed. Afterward, the samples were attached to the center of the vibratome plate using superglue with the frontal side facing upwards. The first sections had a thickness of 250 $\mu$m to reach the region of interest (ROI). After about 3–5 slices, the thickness was adjusted to 120 $\mu$m and cut with a speed of 0.02–0.08 mm/s. After about four sections, vibratome sections could be collected into 1x PBS for furthermore embedding.

## Conventional embedding

The vibratome sections were once more post-fixed overnight in 2% PFA, 2.5% glutaraldehyde in 0.1 M cacodylate buffer (pH 7.2) on ice. On the second day, samples were washed thrice with 0.1 M sodium cacodylate buffer for 10 min each. Afterward, the samples were incubated in 1% $OsO_4$ (75632; Sigma-Aldrich; vol/vol in 0.1 M sodium cacodylate buffer) for 1 h. Then, they were washed twice with cacodylate buffer for 10 min each and afterward three times with $H_2O$ for 5 min each. This was followed by an en bloc staining with 1% uranyl acetate in $H_2O$ for 1 h and three washing steps in $H_2O$ for 10 min each. Samples were dehydrated with increasing ethanol concentrations (5 min 30% ethanol in $H_2O$, 5 min 50% ethanol in $H_2O$, 10 min in 70% ethanol in $H_2O$, 2 × 10 min in 95% ethanol in $H_2O$, 3 × 12 min in 100% ethanol). Finally, the samples were infiltrated in epoxy resin (R1140, AGAR-100; Plano) in 100% ethanol and epoxy resin (1:1 [vol/vol]) for 30 min and another 90 min with a fresh solution. Overnight, they were left in 100% epoxy resin at room temperature on a wheel. On the third day, the epoxy resin was exchanged once. After 6 h, the three regions (the hippocampal CA1 and dentate gyrus [DG] areas, as well as the piriform cortex) were separated. For each animal and age, 3–4 vibratome sections were cut and fixed. Two sections for each animal were furthermore processed by taking out the ROI to embed them individually. The separated regions were finally placed in embedding molds filled with epoxy resin and left for polymerization at 70°C in a drying oven for at least 48 h.

## Trimming, ultrathin sectioning

The blocks were trimmed to the ROI using a file and a razor blade and finally a 45° diamond knife (Diatome) in an EM UC7 ultramicrotome (Leica, Microsystems) to smooth the block face. Using a 35° diamond knife (Diatome), ultrathin sections of 110 nm were obtained and 3–5 ultrathin sections were placed on finder copper grids (G239A-F1; Plano).

## Transmission electron microscopy for nanoSIMS

For the sections required for the correlation of ultrastructure and nanoSIMS, it was necessary to create a high-resolution, large-area image of the different brain ROI. Overview images were taken at a JEM-1011 transmission electron microscope (JEOL) equipped with a Gatan Orius SC1000A camera using the Digital Micrograph software package at different magnifications (Gatan). To cover a large area, a series of contiguous images, which could later be stitched into one image by ImageJ (Schneider et al, 2012), were acquired. For an overview, images at 300x and 800x magnification were taken. Afterward a series of images was acquired manually at a 2,500x magnification. It was essential to provide sufficient overlap of about 20% of the adjacent images. Labeling of the images was performed as described in Preibisch et al (2009).

## Image stitching

ImageJ with the plugin Grid/Collection stitching was used to stitch the images. Once the plugin was selected, the *snake by rows* pattern was selected. The Tile overlap was set to 20%, but could also be less if required. *First file index i*, defines with which number the numbering of the images begins and was selected accordingly. In the directory, the according folder was selected containing the desired images. In the *File names for tiles* section, the image name of the starting image was entered, with all images saved. The number of the sequence had to be replaced with {i}. The Output textfile name remained at TileConfiguration.txt. For the Fusion method, *Linear Blending* was selected. The thresholds remained as the program proposed them for the *Regression threshold* at 0.3, *Max/avg displacement threshold* at 2.5, and *Absolute displacement threshold* at 3.5. In the following list, *Compute overlap* was selected. The *Computation parameters* were set to "*save computation time (but use more RAM)*" and the *Image output* to "*Fuse and display.*" After confirmation, the image was stitched and displayed by ImageJ.

## NanoSIMS imaging

Following EM imaging, the same area was located and imaged by a NanoSIMS 50 liters ion microprobe (Cameca) operating at 8 keV $^{133}Cs^+$ source. Each area of interest was firstly pre-sputtered with a primary ion beam current of 50 pA until a steady-state of secondary ion beam was reached. To have high spatial resolution, secondary ion images were acquired at low primary current of < 0.5 pA. Sample areas of 9 × 9 $\mu$m to 30 × 30 $\mu$m were scanned at 256 × 256, 512 × 512 or 1024 × 1024 pixels with dwell time ranges from 4 to 11 ms per pixel, yielding pixel sizes of between 20–43 nm and mostly below 30 nm. Mass resolving power was adjust to have $^{12}C^{14}N$ and $^{12}C^{15}N$ well separated from their isobaric interferences such as $^{12}C^{13}C^1H$ and $^{13}C^{14}N$, respectively. All figures demonstrated and analyzed represent data from one plane of $^{12}C^{14}N$ and $^{12}C^{15}N$ image taken simultaneously without accumulation.

## Image processing

Each nanoSIMS imaged area was correlated with previously taken EM image in Adobe PhotoshopCS6 (Adobe Systems Incorporated) with high precision. After image registration, a custom-written Matlabmacro (the Mathworks Inc.) was applied to calculate the average $^{15}N/^{14}N$ ratio in each manually chosen ROI as well as in each whole image. Specifically, the selection of ROI was performed on the correlated EM image demonstrating all fine structures in each brain area. In total, data from more than 5,000 ROIs were included in the statistics presented in this text. Each selected ROI yield average $^{15}N/^{14}N$ ratio by dividing its respective $^{12}C^{15}N$ and

$^{12}C^{14}N$ counts. The result for all ROIs from a single nanoSIMS image was saved as a text file, which was furthermore imported in excel for annotation of the structure identity and combine result of all images into one file for furthermore statistical analysis. In this study, mouse brain ultrastructure of presynapse, active zone, postsynapse, PSD, myelin, axon, mitochondria, and neurite are of particular interest and thus selected as ROIs. Note that due to the abundance of mitochondria and neurite in all images, not all of them but a representative fraction was selected from all the imaged areas. HSI image of $^{15}N/^{14}N$ ratio was generated by the OpenMIMS plugin from Fiji (https://github.com/BWHCNI/OpenMIMS). The lower limit for the ratio image was set to 37, which corresponding to natural $^{15}N$ ratio of 0.37%, whereas the upper limit was set to 2,500 for all $^{15}N/^{14}N$ ratio HSI images in different figures.

## Statistics

Statistical box-plots and bar-errorbar plots were generated either in R or Python. Test methods are indicated in each figure caption. Significance levels of all statistical tests in all boxplots and bar-errorbar plots are indicated as the follows: ns: no significant differences between samples, $*P < 0.05$, $**P < 0.01$, $***P < 0.001$, $****P < 0.0001$.

## Supplementary Information

## Acknowledgements

We thank S Langer and K Grewe (University Medical Center Göttingen, Germany) for expert technical assistance. We thank Jasmina Redzovic (University Medical Center Göttingen, Germany) for supporting the initial sample preparation for electron microscopy. We thank Ronja Rehm and Kim-Ann Saal (University Medical Center Göttingen, Germany) for support with mouse-related procedures. SO Rizzoli and C Wichmann were funded by the German Research Foundation (Deutsche Forschungsgemeinschaft, DFG), through the Collaborative Research Center (CRC) 1286, projects A03 and A04. SO Rizzoli and C Wichmann were further supported by the Deutsche Forschungsgemeinschaft under Germany's Excellence Strategy–EXC 2067/1-390729940. SO Rizzoli acknowledges further support from DFG grants RI 1967/10-1 and RI 1967/7-3.

## Author Contributions

F Li: conceptualization, data curation, formal analysis, validation, investigation, methodology, and writing—original draft, review, and editing.
JN Bahr: formal analysis, investigation, methodology, and writing—review and editing.
FA-L Bierth: formal analysis, investigation, methodology, and writing—review and editing.
S Reshetniak: data curation, investigation, visualization, and writing—review and editing.

C Tetzlaff: investigation, methodology, and writing—review and editing.
EF Fornasiero: resources, methodology, and writing—review and editing.
C Wichmann: conceptualization, resources, data curation, software, formal analysis, supervision, funding acquisition, validation, investigation, visualization, methodology, project administration, and writing—original draft, review, and editing.
SO Rizzoli: conceptualization, resources, data curation, software, formal analysis, supervision, funding acquisition, validation, investigation, visualization, methodology, project administration, and writing—original draft, review, and editing.

## Conflict of Interest Statement

The authors declare that they have no conflict of interest.

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
