## [Reviewer comments · Life Science Alliance]

Life Science Alliance

Morphological correlates of synaptic protein turnover in the mouse brain

Fengxia Li, Julius Bahr, Felicitas Bierth, Sofiia Reshetniak, Christian Tetzlaff, Eugenio Fornasiero, Carolin Wichmann, and Silvio Rizzoli

DOI: <https://doi.org/10.26508/lsa.202402793>

Corresponding author(s): Silvio Rizzoli, Universitätsmedizin Göttingen and Carolin Wichmann, Universitätsmedizin Göttingen

Review Timeline:	Submission Date:	2024-04-24
	Editorial Decision:	2024-06-12
	Revision Received:	2024-08-02
	Editorial Decision:	2024-08-02
	Revision Received:	2024-08-05
	Accepted:	2024-08-05

Transaction Report:

June 12, 2024

Re: Life Science Alliance manuscript #LSA-2024-02793-T

Prof. Silvio O Rizzoli
University of Göttingen Medical Center
Department of Neuro- and Sensory Physiology
Humboldtallee 23
Goettingen 37073
Germany

Dear Dr. Rizzoli,

Thank you for submitting your manuscript entitled "Morphological correlates of synaptic protein turnover in the mouse brain" to Life Science Alliance. The manuscript was assessed by expert reviewers, whose comments are appended to this letter. We invite you to submit a revised manuscript addressing the Reviewer comments.

Thank you for this interesting contribution to Life Science Alliance. We are looking forward to receiving your revised manuscript.

Sincerely,

B. MANUSCRIPT ORGANIZATION AND FORMATTING:

Reviewer #1 (Comments to the Authors (Required)):

The rapid turnover of regulatory synaptic proteins at active zones maintain the function and prevent damages at synaptic boutons. However, it is rather unknown how this process relates to synaptic morphology. This paper used nanoscale secondary ion mass spectrometry to detect newly synthesized synaptic components in the brains of young adult and aged mice and fine-scale transmission electron microscopy to investigate synapse morphology. As demonstrated here, several structural parameters correlated to protein turnover, including the size of the pre- and postsynaptic density, the number of synaptic vesicles and the presence of a postsynaptic nascent zone. In contrast to young mice, the turnover of all brain compartments was reduced by ~20% in aged mice. The turnover rates of the pre- and postsynaptic structures correlated well in aged mice, suggesting that they are subject to common regulatory mechanisms. This correlation was poorer in young adult mice, in line with their higher synaptic dynamics. Thus, the authors conclude that synapse turnover is reflected by multiple aspects influencing synaptic morphology.

This is a very nice study about the role and turnover of synaptic proteins at active zones as demonstrated by secondary nanoscale ion mass spectroscopy and high-resolution transmission electron microscopy. The paper is clearly written in all parts, the material and methods section provide sufficient detail to follow the experimental procedure, the results are accompanied by very good figures and finally the results are discussed with the appropriate literature. However, there are some points the reviewer would like to raise. First, what was the reason for the age of the mice. In this study the authors have taken mice 6 month and 24 month in age. It has been shown for example by electron microscopy that no significant differences in structural subelements of synaptic boutons were found, as investigated here, between these mice and adult mice. The reviewer would have expected that the turnover of proteins would be much higher during early development of the brain in much younger mice. Could the authors please comment on that. Secondly, the authors describe a correlation between protein turnover with the size of the presynapse, the pre- and postsynaptic density and the pool of synaptic vesicles which beside the shape and size of the active zone is a key structural parameter in synaptic transmission, strength and efficacy. How was the difference the size of boutons, active zones and in pool size measured? Did the authors look differences in the number of 'docked' vesicles? Please comment on that! Third, was the reason to choose the CA1 and dentate gyrus subregions and not CA3 in the hippocampus?

Minor points:

It would be nice to enlarge the EM pictures of synaptic boutons to provide more detail, for example the shape and size of active zones (see Figure 6) and the pool of synaptic vesicles.

In all the figures representing box plots it is not explained what is represented by the line within the bars, does that represent the mean or median, the vertical lines do they represent the minimum and maximum, does the box represent the interquartile range. The authors should explain this!

In summary, this is a quite nice paper and could be acceptable for publication in this journals after the suggestions raised by the reviewer.

Reviewer #2 (Comments to the Authors (Required)):

This is the first correlative study between TEM and NanoSIMS on turnover of synaptic proteins in intact brains. This study further adds more details in this line of work from this lab, and provides useful and interesting data.

Although this reviewer can follow the manuscript and get the story, I wish the paper is better edited in order for the general readers to understand it easily.

One main suggestion is that measurement criteria need to be specifically defined in the Methods section and then illustrated in one of the EM panels in figures 4-6. For example, the authors mentioned synapse size, which included
1. "Presynapse area", does this mean the area of the presynaptic terminal? How did the authors mark the borders for this area need to be described and illustrated.

2. "Postsynapse area", does this mean the area of the spine? It can easily be marked in one of the EM panels.
3. "PSD area", need to be better defined. Otherwise, it will be mistaken as the "en face" area of the PSD. I know the authors meant the area of the PSD shown in these thin sections. The area need to be marked in at least one panel of the EM images to show how the authors traced the area of the postsynaptic density.
4. "PSD length", ditto here

The rest of comments are mostly minor but need to be corrected:

Abstract

Line 2, "prevent damaged" to "prevent from being damaged" or "prevent damages"

Line 4, specify age of animals : young adult (6 month) and aged (24 month) mice

Is graphic abstract mandatory? I find it too sketchy and not very clear at all.

1. The three regions of the brain (hippocampal CA1, DG and the piriform cortex) should be labeled.
2. The color scheme of Low/high turnover in this diagram is not consistent with the rest of the text, where the pseudocolor scheme used red for high turnover, and blue for low turnover.
3. Are all synapses in the 6 month old young animal with high turnover in presynaptic terminals and low turnover in postsynaptic spines?

Introduction

Line 5 "proteins find themselves" to "proteins reside"

Line 12 "synaptic vesicle delivery" to " delivery of synaptic vesicle proteins"

Results

1st paragraph,

Line 2 "as lysine" to "like lysine"

Line 13 "processed to samples of" to "sectioned to"

Fig 1 legend, it is very important to add "perfusion fixed" before "extracted" and delete the following "fixed". Otherwise, it could be easily misunderstood that brains were extracted and then immersion-fixed.

2nd paragraph, better to specify early on that you are presenting data from young adult here.

For Fig. 2, When citing each specific structure, better to name specific panels in the figure. For example, line 6, myeline should be identified as (zoom2 in Fig. 2A, B), synapses (zoom 1 in Fig. 2A, B), and the intermediate level in mitochondria (zoom 3) should be mentioned in the text. These specific citations will help the general readers to follow the text, and should be applied to all figures.

Fig. 2 legend, the 3 zoomed region need to be identified as synapse, myelin, mitochondria.

Near the end of page 6, is there any EM images of "nascent zone" illustrated?

Page 10, Fig. 7 legend, line 5 (C) to (D)

Methods

P13. "2.500x" should be "2,500x"

Reviewer #3 (Comments to the Authors (Required)):

The paper by Li et al. explores the important question of protein turnover at synapses and its dependence on age. The authors use the powerful combination of electron microscopy followed by nanoSIMS on the same sections to ultrastructurally identify synapses and quantify protein turnover at defined synaptic and other compartments in the neuropil of 3 different areas and 2 ages. The results show high turnover of proteins specifically at synapses, with a 20% reduction at older animals, and define morphological parameters, such as synapse size or the presence of nascent zones, that correlate with protein turnover. These results reveal another crucial aspect of synapse diversity and its relation to synapse morphology, and thus contribute to the growing amount of evidence aimed at enabling functional interpretation of ultrastructural information. Additionally, there is good correlation between turnover in pre- and postsynaptic components in aged mice, as well as in cortex of younger mice,

suggesting that synapse regulation is specific to individual synapses and is not determined exclusively by the pre- and postsynaptic neuron type.

The data overall supports the conclusions well. The Figures and the images are well presented and convincing of the quality of the data. One major drawback is that the analysis is done on individual sections and there is no volume information. This drawback, which is a technological limitation, is acknowledged by the authors and largely taken into account when interpreting the results. While the 2D analysis could obscure some trends (e.g. the size comparisons are effectively between one population of large synapses and another mixed population of small synapses and large synapses sectioned at the edges), it is not expected to affect the main conclusions.

The interpretation and discussion of the results is at times difficult to follow and incomplete. For example, the distinction between "rearrangements of pre-existing synaptic components, rather than the generation of new proteins" that is mentioned in the Discussion, should be introduced much earlier, in the Introduction. Thus, synaptic vesicle delivery to the synapse does not necessarily equate addition of newly generated proteins; synaptic vesicles are known to be shared among adjacent synapses and could redistribute based on activity.

Specific comments:

Introduction, 2nd paragraph: "...this implies that the vesicle pools of small synapses should turn over faster than large synapses". This is true only assuming same levels of activity. Also the whole sentence is difficult to understand.

Introduction, 3rd paragraph, last sentence: synaptic remodeling does not necessarily mean protein turnover as the authors themselves explain much later in the Discussion.

Results, 2nd paragraph: a description of how the different compartments were defined will be helpful. For example, how were the boundaries of a pre or postsynapse defined? If there was a mitochondrion in any compartment, were they included? From the beautiful examples that the authors provide in the figures, it often looks like the mitochondria have different turnover rates from the compartment that they are in. How were neurites defined? Presumably, there were some very large dendritic profiles, but looking at the plots it seems that only smaller neurites were analysed, what was the criteria?

Considering the lateral resolution of nanoSIMS (50-100nm), does it make sense to differentiate between PSD and active zone?

Results, Figure 1: the labels on the color bar in B are very hard to read, please make bigger. Also, how was the 21 day period decided on? A brief mention of a rationale and how this compares to known times of turnover of synaptic proteins will help the reader.

Results, Figure 3C legend: "This analysis indicates that the presynapse turns over especially slowly" - please clarify that this is in the adult. Also, because the effect is rather small, "especially" does not convey this, "the presynapse turns over significantly more slowly" would be a more appropriate.

Results, Morphological parameters section, 2nd paragraph: "larger synapses tending to be older (showing less turnover)". When describing the results, the observation ("showing less turnover") should come first, then followed by the interpretation ("tending to be older").

Results, Figure 6D: will be helpful to indicate which datapoints come from the young vs. aged mice.

Discussion, 2nd paragraph, 1st sentence: not clear.

Discussion, 3rd paragraph: Could the correlation between large postsynaptic size and slow turnover be because large spines tend to be more stable with time?

Also, conversely, the finding that small postsynaptic size is correlated with high turnover, could reflect the combination of two factors: one, that small spines are more dynamic, but also edges of large PSDs on large spines are more dynamic and on single sections they will tend to appear as small postsynapses.

Methods, Vibratome section: "(RO" should be "(ROI)".

Methods, Conventional embedding section: the catalog number for Osmium is misspelled.

Methods, Conventional embedding section: "After 6 hours..": specify the 3 regions.

Reviewer #1 (Comments to the Authors (Required)):

The rapid turnover of regulatory synaptic proteins at active zones maintain the function and prevent damages at synaptic boutons. However, it is rather unknown how this process relates to synaptic morphology. This paper used nanoscale secondary ion mass spectrometry to detect newly synthesized synaptic components in the brains of young adult and aged mice and fine-scale transmission electron microscopy to investigate synapse morphology. As demonstrated here, several structural parameters correlated to protein turnover, including the size of the pre- and postsynaptic density, the number of synaptic vesicles and the presence of a postsynaptic nascent zone. In contrast to young mice mice, the turnover of all brain compartments was reduced by ~20% in aged mice. The turnover rates of the pre- and postsynaptic structures correlated well in aged mice, suggesting that they are subject to common regulatory mechanisms. This correlation was poorer in young adult mice, in line with their higher synaptic dynamics. Thus, the authors conclude that synapse turnover is reflected by multiple aspects influencing synaptic morphology.

This is a very nice study about the role and turnover of synaptic proteins at active zone as demonstrated by secondary nanoscale ion mass spectroscopy and high-resolution transmission electron microscopy.

The paper is clearly written in all parts, the material and methods section provide sufficient detail to follow the experimental procedure, the results are accompanied by very good figures and finally the results are discussed with the appropriate literature.

We thank the Reviewer for the comments.

However, there are some points the reviewer would like to raise. First, what was the reason for the age of the mice. In this study the authors have taken mice 6 month and 24 month in age. It has been shown for example by electron microscopy that no significant differences in structural subelements of synaptic boutons were found, as investigated here, between these mice and adult mice. The reviewer would have expected that the turnover of proteins would be much higher during early development of the brain in much younger mice. Could the authors please comment on that.

According to the life history stages of the C57BL/6J mice (adapted from Flurkey *et al.*, 2007), the physiological age of a 6-month old mouse is comparable to that of a 30-year old human, indicating mature adulthood, but not yet a middle-age setting. Younger mice may have juvenile traits, as ongoing myelination, and were therefore avoided. The 24-month age is similar to 69 years in the human, and would be representative of old age. Older mice tend to have poor physical health, which would have complicated our analysis.

Secondly, the authors describe a correlation between protein turnover with the size of the presynapse, the pre- and postsynaptic density and the pool of synaptic vesicles which beside the shape and size of the active zone is a key structural parameter in synaptic transmission, strength and efficacy. How was the difference the size of boutons, active zones and in pool size measured?

All analyses were performed on the 2D-electron microscopy images. The different structures were traced by hand, on these images, and the respective turnover values were then measured in the corresponding nanoSIMS images. To determine the vesicle pool sizes, all vesicles were counted within the respective electron microscopy images. We have now added a supplementary figure (Fig. S1), in which we explain the measurements (the drawing procedure).

Did the authors looked differences in the number of 'docked' vesicles? Please comment on that!

We thank the Reviewer for this important comment. Indeed, the number of docked vesicles could make a difference, since synapses in which docked vesicles are lacking in specific regions, opposite the PSD (nascent zones) tend to show a higher turnover level for the PSD (Fig. 6D).

We have now measured this in more detail, by performing an analysis of the correlation of turnover to the number of docked vesicles. We could not find any significant correlations, as now noted on page 4.

Third, was the reason to choose the CA1 and dentate gyrus subregions and not CA3 in the hippocampus?

The dentate gyrus was selected as it is a brain region known for relatively frequent neurogenesis, which could, therefore, exhibit a different turnover behavior to other regions. This was not the case, as shown in our manuscript.

The CA1 area was chosen, rather than the CA3 area, since it is less strongly connected to the dentate gyrus. We wanted to avoid regions that are strongly connected to each other, to avoid having a functional bias in our analysis of different regions. In principle, regions that work very closely together could show similar turnover phenotypes, which may bias an analysis such as ours.

Minor points:

It would be nice to enlarge the EM pictures of synaptic boutons to provide more detail, for example the shape and size of active zones (see Figure 6) and the pool of synaptic vesicles.

We have increased the size of the EM images.

In all the figures representing box plots it is not explained what is represented by the line within the bars, does that represent the mean or median, the vertical lines do they represent the minimum and maximum, does the box represent the interquartile range. The authors should explain this!

We now include this information in the legend of the respective figures. The box plots show the median (middle line), the lower and upper quartiles (box), the minimum and maximum numbers that are not outliers (the whiskers), and the outliers (calculated using the interquartile range).

In summary, this is a quite nice paper and could be acceptable for publication in this journals after the suggestions raised by the reviewer.

We thank the Reviewer for the comments.

Reviewer #2 (Comments to the Authors (Required)):

This is the first correlative study between TEM and NanoSIMS on turnover of synaptic proteins in intact brains. This study further adds more details in this line of work from this lab, and provides useful and interesting data.

Although this reviewer can follow the manuscript and get the story, I wish the paper is better edited in order for the general readers to understand it easily.

We thank the Reviewer for the comments. We adjusted the manuscript as indicated by the Reviewer.

One main suggestion is that measurement criteria need to be specifically defined in the Methods section and then illustrated in one of the EM panels in figures 4-6.

We now include a supplementary figure to explain this. Please see the new Fig. S1.

For example, the authors mentioned synapse size, which included

1. "Presynapse area", does this mean the area of the presynaptic terminal? How did the authors mark the borders for this area need to be described and illustrated.

2. "Postsynapse area", does this mean the area of the spine? It can easily be marked in one of the EM panels.

In both cases, area means the two-dimensional area of the respective pre- or postsynapses. The values were obtained by drawing onto the respective EM images, as shown in Fig. S1.

3. "PSD area", need to be better defined. Otherwise, it will be mistaken as the "en face" area of the PSD.

I know the authors meant the area of the PSD shown in these thin sections. The area need to be marked in at least one panel of the EM images to show how the authors traced the area of the postsynaptic density.

4. "PSD length", ditto here

We have now explained these terms in the text, especially in the legend of Fig. 6, and also in Fig. S1.

The rest of comments are mostly minor but need to be corrected:

Abstract

Line 2, "prevent damaged" to "prevent from being damaged" or "prevent damages"

Line 4, specify age of animals : young adult (6 month) and aged (24 month) mice

We corrected these issues.

Is graphic abstract mandatory? I find it too sketchy and not very clear at all.

1. The three regions of the brain (hippocampal CA1, DG and the piriform cortex) should be labeled.

2. The color scheme of Low/high turnover in this diagram is not consistent with the rest of the text, where the pseudocolor scheme used red for high turnover, and blue for low turnover.

3. Are all synapses in the 6 month old young animal with high turnover in presynaptic terminals and low turnover in postsynaptic spines?

The graphic abstract is optional. We agree with the Reviewer that it is very difficult to capture all of our points in a graphical abstract, and that some aspects, such as the variability in the turnover levels in the young animals, are too complex to be presented in this fashion. We have therefore removed the graphic abstract.

Introduction

Line 5 "proteins find themselves" to "proteins reside"

Line 12 "synaptic vesicle delivery" to " delivery of synaptic vesicle proteins"

Results

1st paragraph, Line 2 "as lysine" to "like lysine"

Line 13 "processed to samples of" to "sectioned to"

Fig 1 legend, it is very important to add "perfusion fixed" before "extracted" and delete the following "fixed". Otherwise, it could be easily misunderstood that brains were extracted and then immersion-fixed.

2nd paragraph, better to specify early on that you are presenting data from young adult here.

For Fig. 2, When citing each specific structure, better to name specific panels in the figure. For example, line 6, myeline should be identified as (zoom2 in Fig. 2A, B), synapses (zoom 1 in Fig. 2A, B), and the intermediate level in mitochondria (zoom 3) should be mentioned in the text. These specific citations will help the general readers to follow the text, and should be applied to all figures.

Fig. 2 legend, the 3 zoomed region need to be identified as synapse, myelin, mitochondria.

We have now addressed all of these points.

Near the end of page 6, is there any EM images of "nascent zone" illustrated?

We now indicate a few nascent zones in Fig. 6 with asterisks.

Page 10, Fig. 7 legend, ling 5 (C) to (D)

Methods

P13. "2.500x" should be "2,500x"

We have now addressed all of these points.

Reviewer #3 (Comments to the Authors (Required)):

The paper by Li et al. explores the important question of protein turnover at synapses and its dependence on age. The authors use the powerful combination of electron microscopy followed by nanoSIMS on the same sections to ultrastructurally identify synapses and quantify protein turnover at defined synaptic and other compartments in the neuropil of 3 different areas and 2 ages. The results show high turnover of proteins specifically at synapses, with a 20% reduction at older animals, and define morphological parameters, such as synapse size or the presence of nascent zones, that correlate with protein turnover. These results reveal another crucial aspect of synapse diversity and its relation to synapse morphology, and thus contribute to the growing amount of evidence aimed at enabling functional interpretation of ultrastructural information. Additionally, there is good correlation between turnover in pre- and postsynaptic components in aged mice, as well as in cortex of younger mice, suggesting that synapse regulation is specific to individual synapses and is not determined exclusively by the pre- and postsynaptic neuron type.

The data overall supports the conclusions well. The Figures and the images are well presented and convincing of the quality of the data. One major drawback is that the analysis is done on individual sections and there is no volume information. This drawback, which is a technological limitation, is acknowledged by the authors and largely taken into account when interpreting the results. While the 2D analysis could obscure some trends (e.g. the size comparisons are effectively between one population of large synapses and another mixed population of small synapses and large synapses sectioned at the edges), it is not expected to affect the main conclusions.

We thank the Reviewer for the comments.

The interpretation and discussion of the results is at times difficult to follow and incomplete. For example, the distinction between "rearrangements of pre-existing synaptic components, rather than the generation of new proteins" that is mentioned in the Discussion, should be introduced much earlier, in the Introduction. Thus, synaptic vesicle delivery to the synapse does not necessarily equate addition of newly generated proteins; synaptic vesicles are known to be shared among adjacent synapses and could redistribute based on activity.

We have now addressed this point on page 2.

Specific comments:

Introduction, 2nd paragraph: "...this implies that the vesicle pools of small synapses should turn over faster than large synapses". This is true only assuming same levels of activity. Also the whole sentence is difficult to understand.

We have now corrected the respective phrases.

Introduction, 3rd paragraph, last sentence: synaptic remodeling does not necessarily mean protein turnover as the authors themselves explain much later in the Discussion.

We now indicate this clearly in the introduction.

Results, 2nd paragraph: a description of how the different compartments were defined will be helpful. For example, how were the boundaries of a pre or postsynapse defined? If there was a mitochondrion in any compartment, were they included? From the beautiful examples that the authors provide in the figures, it often looks like the mitochondria have different turnover rates from the compartment that they are in. How were neurites defined? Presumably, there were some very large dendritic profiles, but looking at the plots it seems that only smaller neurites were analysed, what was the criteria?

We now present this in a supplementary figure (Fig. S1). The Reviewer is right in pointing out that mitochondria show a different turnover to the compartments in which they find themselves, as we also found in bulk mass spectrometry experiments in the past (Fornasiero et al., 2018). We avoided

including the mitochondria in the analysis of the respective compartments, and measured them separately. Neurites were defined as small, round profiles, whose morphology did not indicate a specific cellular identity. They were only used as a control, a “random cellular component”, whose morphology was not expected to correlate to their turnover.

Considering the lateral resolution of nanoSIMS (50-100nm), does it make sense to differentiate between PSD and active zone?

The Reviewer is right in pointing out that the two components are very close to each other, implying that some degree of signal imprecision is expected. The higher protein levels in the PSD (visible in all of our images) may influence the active zone measurement. Nonetheless, we performed numerous measurements in areas in which only one of these components was visible, due to the geometry of the EM sectioning, ensuring that we can, indeed, report such values.

Results, Figure 1: the labels on the color bar in B are very hard to read, please make bigger.

We made this change.

Also, how was the 21 day period decided on? A brief mention of a rationale and how this compares to known times of turnover of synaptic proteins will help the reader.

The 21-day period was determined empirically, following our work from bulk analyses of protein turnover in the brain (Fornasiero et al., 2018). During the first two weeks of feeding, turnover is relatively rapid, and the amino acid populations change rapidly between the ones obtained from the diet and those derived from the degradation of the body's own proteins. At 21 days, the system is at steady-state, implying that lower variation will be found between animals. We now note this in the Results section.

Results, Figure 3C legend: "This analysis indicates that the presynapse turns over especially slowly" - please clarify that this is in the adult. Also, because the effect is rather small, "especially" does not convey this, "the presynapse turns over significantly more slowly" would be a more appropriate.

We made these corrections.

Results, Morphological parameters section, 2nd paragraph: "larger synapses tending to be older (showing less turnover)". When describing the results, the observation ("showing less turnover") should come first, then followed by the interpretation ("tending to be older").

We made this correction.

Results, Figure 6D: will be helpful to indicate which datapoints come from the young vs. aged mice.

We now show this in a new Fig. S10.

Discussion, 2nd paragraph, 1st sentence: not clear.

We have now clarified the respective phrase.

Discussion, 3rd paragraph: Could the correlation between large postsynaptic size and slow turnover be because large spines tend to be more stable with time?

That is indeed our observation, that large spines tend to be metabolically more stable. We are unsure whether these synapses are also more stable in terms of physiology, being subjected to less remodeling, since our experiments cannot test this issue. We have added this hypothesis to the manuscript.

Also, conversely, the finding that small postsynaptic size is correlated with high turnover, could reflect the combination of two factors: one, that small spines are more dynamic, but also edges of large PSDs

on large spines are more dynamic and on single sections they will tend to appear as small postsynapses.

We have added this hypothesis to the manuscript.

Methods, Vibratome section: "(RO" should be "(ROI)".

We made this correction.

*Methods, Conventional embedding section: the catalog number for Osmium is misspelled.
Methods, Conventional embedding section: "After 6 hours..": specify the 3 regions.*

We made these corrections.

References

Flurkey K, Curren J & Harrison D (2007). Mouse models in aging research. *Faculty Research* 2000 - 2009 637–672.

Fornasiero EF et al. (2018). Precisely measured protein lifetimes in the mouse brain reveal differences across tissues and subcellular fractions. *Nat Commun* **9**, 4230.

August 2, 2024

RE: Life Science Alliance Manuscript #LSA-2024-02793-TR

Prof. Silvio O Rizzoli
Universitätsmedizin Göttingen
Department of Neuro- and Sensory Physiology
Humboldtallee 23
Goettingen 37073
Germany

Dear Dr. Rizzoli,

Thank you for submitting your revised manuscript entitled "Morphological correlates of synaptic protein turnover in the mouse brain". We would be happy to publish your paper in Life Science Alliance pending final revisions necessary to meet our formatting guidelines.

- please be sure that the authorship listing and order is correct
- please add a Category for your manuscript in our system
- please move your main and supplementary figure legends to the main manuscript text after the References section
- please add an Author Contributions section to your main manuscript text
- please add a Conflict of Interest statement to your main manuscript text
- please add callouts for Figures 1A-B; 3A-B; 4A; 5A; 6A; 7A-D; S2A-B; S3A-B; S4A-B; S5A-B; S6A-C; S7A-B; S8A-B; S9A-B; S10 and S11A-C to your main manuscript text

LSA now encourages authors to provide a 30-60 second video where the study is briefly explained. We will use these videos on social media to promote the published paper and the presenting author (for examples, see <https://docs.google.com/document/d/1-UWCfbE4pGcDdcgzcmiuJl2XMBJnxKYeqRvLLrLSo8s/edit?usp=sharing>). Corresponding or first-authors are welcome to submit the video. Please submit only one video per manuscript. The video can be emailed to contact@life-science-alliance.org

A. FINAL FILES:

B. MANUSCRIPT ORGANIZATION AND FORMATTING:

Sincerely,

August 5, 2024

RE: Life Science Alliance Manuscript #LSA-2024-02793-TRR

Prof. Silvio O Rizzoli
Universitätsmedizin Göttingen
Department of Neuro- and Sensory Physiology
Humboldtallee 23
Goettingen 37073
Germany

Dear Dr. Rizzoli,

Thank you for submitting your Research Article entitled "Morphological correlates of synaptic protein turnover in the mouse brain". It is a pleasure to let you know that your manuscript is now accepted for publication in Life Science Alliance. Congratulations on this interesting work.

DISTRIBUTION OF MATERIALS:

Again, congratulations on a very nice paper. I hope you found the review process to be constructive and are pleased with how the manuscript was handled editorially. We look forward to future exciting submissions from your lab.

Sincerely,
